# The Nearest Neighbor Information Estimator is Adaptively Near Minimax Rate-Optimal

**Jiantao Jiao**
Department of Electrical Engineering and Computer Sciences
University of California, Berkeley
jiantao@berkeley.edu

**Weihao Gao**
Department of ECE
Coordinated Science Laboratory
University of Illinois at Urbana-Champaign
wgao9@illinois.edu

**Yanjun Han**
Department of Electrical Engineering
Stanford University
yjhan@stanford.edu

## Abstract

We analyze the Kozachenko–Leonenko (KL) fixed $k$-nearest neighbor estimator for the differential entropy. We obtain the first uniform upper bound on its performance for any fixed $k$ over Hölder balls on a torus without assuming any conditions on how close the density could be from zero. Accompanying a recent minimax lower bound over the Hölder ball, we show that the KL estimator for any fixed $k$ is achieving the minimax rates up to logarithmic factors without cognizance of the smoothness parameter $s$ of the Hölder ball for $s \in (0, 2]$ and arbitrary dimension $d$, rendering it the first estimator that provably satisfies this property.

## 1 Introduction

Information theoretic measures such as entropy, Kullback-Leibler divergence and mutual information quantify the amount of information among random variables. They have many applications in modern machine learning tasks, such as classification [48], clustering [46, 58, 10, 41] and feature selection [1, 17]. Information theoretic measures and their variants can also be applied in several data science domains such as causal inference [18], sociology [49] and computational biology [36]. Estimating information theoretic measures from data is a crucial sub-routine in the aforementioned applications and has attracted much interest in statistics community. In this paper, we study the problem of estimating Shannon differential entropy, which is the basis of estimating other information theoretic measures for continuous random variables.

Suppose we observe $n$ independent identically distributed random vectors $\mathbf{X} = \{X_1, \ldots, X_n\}$ drawn from density function $f$ where $X_i \in \mathbb{R}^d$. We consider the problem of estimating the differential entropy

$$h(f) \quad = -\int f(x) \ln f(x) dx \,, \tag{1}$$

from the empirical observations $\mathbf{X}$. The fundamental limit of estimating the differential entropy is given by the minimax risk

$$\inf_{\hat{h}} \sup_{f \in \mathcal{F}} \left( \mathbb{E}(\hat{h}(\mathbf{X}) - h(f))^2 \right)^{1/2}, \tag{2}$$

where the infimum is taken over all estimators $\hat{h}$ that is a function of the empirical data $\mathbf{X}$. Here $\mathcal{F}$ denotes a (nonparametric) class of density functions.

The problem of differential entropy estimation has been investigated extensively in the literature. As discussed in [2], there exist two main approaches, where one is based on kernel density estimators [30], and the other is based on the nearest neighbor methods [56, 53, 52, 11, 3], which is pioneered by the work of [33].

The problem of differential entropy estimation lies in the general problem of estimating nonparametric functionals. Unlike the parametric counterparts, the problem of estimating nonparametric functionals is challenging even for smooth functionals. Initial efforts have focused on inference of linear, quadratic, and cubic functionals in Gaussian white noise and density models and have laid the foundation for the ensuing research. We do not attempt to survey the extensive literature in this area, but instead refer to the interested reader to, e.g., [24, 5, 12, 16, 6, 32, 37, 47, 8, 9, 54] and the references therein. For non-smooth functionals such as entropy, there is some recent progress [38, 26, 27] on designing theoretically minimax optimal estimators, while these estimators typically require the knowledge of the smoothness parameters, and the practical performances of these estimators are not yet known.

The $k$-nearest neighbor differential entropy estimator, or Kozachenko-Leonenko (KL) estimator is computed in the following way. Let $R_{i,k}$ be the distance between $X_i$ and its $k$-nearest neighbor among $\{X_1, \ldots, X_{i-1}, X_{i+1}, \ldots, X_n\}$. Precisely, $R_{i,k}$ equals the $k$-th smallest number in the list $\{\|X_i - X_j\| : j \neq i, j \in [n]\}$, here $[n] = \{1, 2, \ldots, n\}$. Let $B(x, \rho)$ denote the closed $\ell_2$ ball centered at $x$ of radius $\rho$ and $\lambda$ be the Lebesgue measure on $\mathbb{R}^d$. The KL differential entropy estimator is defined as

$$\hat{h}_{n,k}(\mathbf{X}) = \ln k - \psi(k) + \frac{1}{n} \sum_{i=1}^{n} \ln \left( \frac{n}{k} \lambda(B(X_i, R_{i,k})) \right), \tag{3}$$

where $\psi(x)$ is the digamma function with $\psi(1) = -\gamma$, $\gamma = -\int_0^\infty e^{-t} \ln t \, dt = 0.5772156\ldots$ is the Euler–Mascheroni constant.

There exists an intuitive explanation behind the construction of the KL differential entropy estimator. Writing informally, we have

$$h(f) = \mathbb{E}_f[-\ln f(X)] \approx \frac{1}{n} \sum_{i=1}^{n} -\ln f(X_i) \approx \frac{1}{n} \sum_{i=1}^{n} -\ln \hat{f}(X_i), \tag{4}$$

where the first approximation is based on the law of large numbers, and in the second approximation we have replaced $f$ by a nearest neighbor density estimator $\hat{f}$. The nearest neighbor density estimator $\hat{f}(X_i)$ follows from the "intuition" [1] that

$$\hat{f}(X_i)\lambda(B(X_i, R_{i,k})) \approx \frac{k}{n}. \tag{5}$$

Here the final additive bias correction term $\ln k - \psi(k)$ follows from a detailed analysis of the bias of the KL estimator, which will become apparent later.

We focus on the regime where $k$ is a fixed: in other words, it does not grow as the number of samples $n$ increases. The fixed $k$ version of the KL estimator is widely applied in practice and enjoys smaller computational complexity, see [52].

There exists extensive literature on the analysis of the KL differential entropy estimator, which we refer to [4] for a recent survey. One of the major difficulties in analyzing the KL estimator is that the nearest neighbor density estimator exhibits a huge bias when the density is small. Indeed, it was shown in [42] that the bias of the nearest neighbor density estimator in fact does not vanish even

when $n \to \infty$ and deteriorates as $f(x)$ gets close to zero. In the literature, a large collection of work assume that the density is uniformly bounded away from zero [23, 29, 57, 30, 53], while others put various assumptions quantifying on average how close the density is to zero [25, 40, 56, 14, 20, 52, 11]. In this paper, we focus on removing assumptions on how close the density is to zero.

## 1.1 Main Contribution

Let $\mathcal{H}_d^s(L; [0, 1]^d)$ be the Hölder ball in the unit cube (torus) (formally defined later in Definition 2 in Appendix A) and $s \in (0, 2]$ is the Hölder smoothness parameter. Then, the worst case risk of the fixed $k$-nearest neighbor differential entropy estimator over $\mathcal{H}_d^s(L; [0, 1]^d)$ is controlled by the following theorem.

**Theorem 1** *Let $\mathbf{X} = \{X_1, \ldots, X_n\}$ be i.i.d. samples from density function $f$. Then, for $0 < s \leq 2$, the fixed $k$-nearest neighbor KL differential entropy estimator $\hat{h}_{n,k}$ in (3) satisfies*

$$\left( \sup_{f \in \mathcal{H}_d^s(L; [0,1]^d)} \mathbb{E}_f \left( \hat{h}_{n,k}(\mathbf{X}) - h(f) \right)^2 \right)^{\frac{1}{2}} \leq C \left( n^{-\frac{s}{s+d}} \ln(n+1) + n^{-\frac{1}{2}} \right). \quad (6)$$

*where $C$ is a constant depends only on $s, L, k$ and $d$.*

The KL estimator is in fact nearly minimax up to logarithmic factors, as shown in the following result from [26].

**Theorem 2** *[26] Let $\mathbf{X} = \{X_1, \ldots, X_n\}$ be i.i.d. samples from density function $f$. Then, there exists a constant $L_0$ depending on $s, d$ only such that for all $L \geq L_0, s > 0$,*

$$\left( \inf_{\hat{h}} \sup_{f \in \mathcal{H}_d^s(L; [0,1]^d)} \mathbb{E}_f \left( \hat{h}(\mathbf{X}) - h(f) \right)^2 \right)^{\frac{1}{2}} \geq c \left( n^{-\frac{s}{s+d}} (\ln(n+1))^{-\frac{s+2d}{s+d}} + n^{-\frac{1}{2}} \right). \quad (7)$$

*where $c$ is a constant depends only on $s$, $L$ and $d$.*

**Remark 1** *We emphasize that one cannot remove the condition $L \geq L_0$ in Theorem 2. Indeed, if the Hölder ball has a too small width, then the density itself is bounded away from zero, which makes the differential entropy a smooth functional, with minimax rates $n^{-\frac{4s}{4s+d}} + n^{-1/2}$ [51, 50, 43].*

Theorem 1 and 2 imply that for any fixed $k$, the KL estimator achieves the minimax rates up to logarithmic factors without knowing $s$ for all $s \in (0, 2]$, which implies that it is near minimax rate-optimal (within logarithmic factors) when the dimension $d \leq 2$. We cannot expect the vanilla version of the KL estimator to adapt to higher order of smoothness since the nearest neighbor density estimator can be viewed as a variable width kernel density estimator with the box kernel, and it is well known in the literature (see, e.g., [55, Chapter 1]) that any positive kernel cannot exploit the smoothness $s > 2$. We refer to [26] for a more detailed discussion on this difficulty and potential solutions. The Jackknife idea, such as the one presented in [11, 3] might be useful for adapting to $s > 2$.

The significance of our work is multi-folded:

- We obtain the first uniform upper bound on the performance of the fixed $k$-nearest neighbor KL differential entropy estimator over Hölder balls without assuming how close the density could be from zero. We emphasize that assuming conditions of this type, such as the density is bounded away from zero, could make the problem significantly easier. For example, if the density $f$ is assumed to satisfy $f(x) \geq c$ for some constant $c > 0$, then the differential entropy becomes a *smooth* functional and consequently, the general technique for estimating smooth nonparametric functionals [51, 50, 43] can be directly applied here to achieve the minimax rates $n^{-\frac{4s}{4s+d}} + n^{-1/2}$. The main technical tools that enabled us to remove the conditions on how close the density could be from zero are the Besicovitch covering lemma (Lemma. 4) and the generalized Hardy–Littlewood maximal inequality.

- We show that, for any fixed $k$, the $k$-nearest neighbor KL entropy estimator nearly achieves the minimax rates without knowing the smoothness parameter $s$. In the functional estimation literature, designing estimators that can be theoretically proved to adapt to unknown

levels of smoothness is usually achieved using the Lepski method [39, 22, 45, 44, 27], which is not known to be performing well in general in practice. On the other hand, a simple plug-in approach can achieves the rate of $n^{-s/(s+d)}$, but only when $s$ is known [26]. The KL estimator is well known to exhibit excellent empirical performance, but existing theory has not yet demonstrated its near-"optimality" when the smoothness parameter $s$ is not known. Recent works [3, 52, 11] analyzed the performance of the KL estimator under various assumptions on how close the density could be to zero, with no matching lower bound up to logarithmic factors in general. Our work makes a step towards closing this gap and provides a theoretical explanation for the wide usage of the KL estimator in practice.

The rest of the paper is organized as follows. Section 2 is dedicated to the proof of Theorem 1. We discuss some future directions in Section 3.

## 1.2 Notations

For positive sequences $a_\gamma, b_\gamma$, we use the notation $a_\gamma \lesssim_\alpha b_\gamma$ to denote that there exists a universal constant $C$ that only depends on $\alpha$ such that $\sup_\gamma \frac{a_\gamma}{b_\gamma} \leq C$, and $a_\gamma \gtrsim_\alpha b_\gamma$ is equivalent to $b_\gamma \lesssim_\alpha a_\gamma$. Notation $a_\gamma \asymp_\alpha b_\gamma$ is equivalent to $a_\gamma \lesssim_\alpha b_\gamma$ and $b_\gamma \lesssim_\alpha a_\gamma$. We write $a_\gamma \lesssim b_\gamma$ if the constant is universal and does not depend on any parameters. Notation $a_\gamma \gg b_\gamma$ means that $\liminf_\gamma \frac{a_\gamma}{b_\gamma} = \infty$, and $a_\gamma \ll b_\gamma$ is equivalent to $b_\gamma \gg a_\gamma$. We write $a \wedge b = \min\{a, b\}$ and $a \vee b = \max\{a, b\}$.

## 2 Proof of Theorem 1

In this section, we will prove that

$$\left( \mathbb{E} \left( \hat{h}_{n,k}(\mathbf{X}) - h(f) \right)^2 \right)^{\frac{1}{2}} \lesssim_{s,L,d,k} n^{-\frac{s}{s+d}} \ln(n+1) + n^{-\frac{1}{2}} , \tag{8}$$

for any $f \in \mathcal{H}_d^s(L; [0,1]^d)$ and $s \in (0, 2]$. The proof consists two parts: (i) the upper bound of the bias in the form of $O_{s,L,d,k}(n^{-s/(s+d)} \ln(n+1))$; (ii) the upper bound of the variance is $O_{s,L,d,k}(n^{-1})$. Below we show the bias proof and relegate the variance proof to Appendix B.

First, we introduce the following notation

$$f_t(x) = \frac{\mu(B(x,t))}{\lambda(B(x,t))} = \frac{1}{V_d t^d} \int_{u:|u-x| \leq t} f(u) du . \tag{9}$$

Here $\mu$ is the probability measure specified by density function $f$ on the torus, $\lambda$ is the Lebesgue measure on $\mathbb{R}^d$, and $V_d = \pi^{d/2}/\Gamma(1+d/2)$ is the Lebesgue measure of the unit ball in $d$-dimensional Euclidean space. Hence $f_t(x)$ is the average density of a neighborhood near $x$. We first state two main lemmas about $f_t(x)$ which will be used later in the proof.

**Lemma 1** *If $f \in \mathcal{H}_d^s(L; [0,1]^d)$ for some $0 < s \leq 2$, then for any $x \in [0,1]^d$ and $t > 0$, we have*

$$|f_t(x) - f(x)| \leq \frac{dLt^s}{s+d} , \tag{10}$$

**Lemma 2** *If $f \in \mathcal{H}_d^s(L; [0,1]^d)$ for some $0 < s \leq 2$ and $f(x) \geq 0$ for all $x \in [0,1]^d$, then for any $x$ and any $t > 0$, we have*

$$f(x) \lesssim_{s,L,d} \max \left\{ f_t(x), \left( f_t(x) V_d t^d \right)^{s/(s+d)} \right\} , \tag{11}$$

*Furthermore, $f(x) \lesssim_{s,L,d} 1$.*

We relegate the proof of Lemma 1 and Lemma 2 to Appendix C. Now we investigate the bias of $\hat{h}_{n,k}(\mathbf{X})$. The following argument reduces the bias analysis of $\hat{h}_{n,k}(\mathbf{X})$ to a function analytic problem. For notation simplicity, we introduce a new random variable $X \sim f$ independent of

$\{X_1, \ldots, X_n\}$ and study $\hat{h}_{n+1,k}(\{X_1, \ldots, X_n, X\})$. For every $x \in \mathbb{R}^d$, denote $R_k(x)$ by the $k$-nearest neighbor distance from $x$ to $\{X_1, X_2, \ldots, X_n\}$ under distance $d(x, y) = \min_{m \in \mathbb{Z}^d} \|m + x - y\|$, i.e., the $k$-nearest neighbor distance on the torus. Then,

$$\mathbb{E}[\hat{h}_{n+1,k}(\{X_1, \ldots, X_n, X\})] - h(f) \tag{12}$$

$$= -\psi(k) + \mathbb{E}\left[\ln\left((n+1)\lambda(B(X, R_k(X)))\right)\right] + \mathbb{E}\left[\ln f(X)\right] \tag{13}$$

$$= \mathbb{E}\left[\ln\left(\frac{f(X)\lambda(B(X, R_k(X)))}{\mu(B(X, R_k(X)))}\right)\right] + \mathbb{E}\left[\ln\left((n+1)\mu(B(X, R_k(X)))\right)\right] - \psi(k) \tag{14}$$

$$= \mathbb{E}\left[\ln\frac{f(X)}{f_{R_k(X)}(X)}\right] + \left(\mathbb{E}\left[\ln\left((n+1)\mu(B(X, R_k(X)))\right)\right] - \psi(k)\right). \tag{15}$$

We first show that the second term $\mathbb{E}\left[\ln\left((n+1)\mu(B(X, R_k(X)))\right)\right] - \psi(k)$ can be universally controlled regardless of the smoothness of $f$. Indeed, the random variable $\mu(B(X, R_k(X))) \sim \text{Beta}(k, n+1-k)$ [4, Chap. 1.2] and it was shown in [4, Theorem 7.2] that there exists a universal constant $C > 0$ such that

$$\left|\mathbb{E}\left[\ln\left((n+1)\mu(B(X, R_k(X)))\right)\right] - \psi(k)\right| \leq \frac{C}{n}. \tag{16}$$

Hence, it suffices to show that for $0 < s \leq 2$,

$$\left|\mathbb{E}\left[\ln\frac{f(X)}{f_{R_k(X)}(X)}\right]\right| \lesssim_{s,L,d,k} n^{-\frac{s}{s+d}}\ln(n+1). \tag{17}$$

We split our analysis into two parts. Section 2.1 shows that $\mathbb{E}\left[\ln\frac{f_{R_k(X)}(X)}{f(X)}\right] \lesssim_{s,L,d,k} n^{-\frac{s}{s+d}}$ and Section 2.2 shows that $\mathbb{E}\left[\ln\frac{f(X)}{f_{R_k(X)}(X)}\right] \lesssim_{s,L,d,k} n^{-\frac{s}{s+d}}\ln(n+1)$, which completes the proof.

## 2.1 Upper bound on $\mathbb{E}\left[\ln\frac{f_{R_k(X)}(X)}{f(X)}\right]$

By the fact that $\ln y \leq y - 1$ for any $y > 0$, we have

$$\mathbb{E}\left[\ln\frac{f_{R_k(X)}(X)}{f(X)}\right] \leq \mathbb{E}\left[\frac{f_{R_k(X)}(X) - f(X)}{f(X)}\right] \tag{18}$$

$$= \int_{[0,1]^d \cap \{x: f(x) \neq 0\}} \left(\mathbb{E}[f_{R_k(x)}(x)] - f(x)\right) dx. \tag{19}$$

Here the expectation is taken with respect to the randomness in $R_k(x) = \min_{1 \leq i \leq n, m \in \mathbb{Z}^d} \|m + X_i - x\|, x \in \mathbb{R}^d$. Define function $g(x; f, n)$ as

$$g(x; f, n) = \sup\left\{u \geq 0 : V_d u^d f_u(x) \leq \frac{1}{n}\right\}, \tag{20}$$

$g(x; f, n)$ intuitively means the distance $R$ such that the probability mass $\mu(B(x, R))$ within $R$ is $1/n$. Then for any $x \in [0, 1]^d$, we can split $\mathbb{E}[f_{R_k(x)}(x)] - f(x)$ into three terms as

$$\mathbb{E}[f_{R_k(x)}(x)] - f(x) = \mathbb{E}[(f_{R_k(x)}(x) - f(x))\mathbb{1}(R_k(x) \leq n^{-1/(s+d)})] \tag{21}$$

$$+ \mathbb{E}[(f_{R_k(x)}(x) - f(x))\mathbb{1}(n^{-1/(s+d)} < R_k(x) \leq g(x; f, n))] \tag{22}$$

$$+ \mathbb{E}[(f_{R_k(x)}(x) - f(x))\mathbb{1}(R_k(x) > g(x; f, n) \vee n^{-1/(s+d)})] \tag{23}$$

$$= C_1 + C_2 + C_3. \tag{24}$$

Now we handle three terms separately. Our goal is to show that for every $x \in [0, 1]$, $C_i \lesssim_{s,L,d} n^{-s/(s+d)}$ for $i \in \{1, 2, 3\}$. Then, taking the integral with respect to $x$ leads to the desired bound.

1. Term $C_1$: whenever $R_k(x) \leq n^{-1/(s+d)}$, by Lemma 1, we have

$$|f_{R_k(x)}(x) - f(x)| \leq \frac{dLR_k(x)^s}{s+d} \lesssim_{s,L,d} n^{-s/(s+d)}, \tag{25}$$

which implies that

$$C_1 \leq \mathbb{E}\left[|f_{R_k(x)}(x) - f(x)|\mathbb{1}(R_k(x) \leq n^{-1/(s+d)})\right] \lesssim_{s,L,d} n^{-s/(s+d)}. \tag{26}$$

2. Term $C_2$: whenever $R_k(x)$ satisfies that $n^{-1/(s+d)} < R_k(x) \le g(x; f, n)$, by definition of $g(x; f, n)$, we have $V_d R_k(x)^d f_{R_k(x)}(x) \le \frac{1}{n}$, which implies that

$$f_{R_k(x)}(x) \le \frac{1}{n V_d R_k(x)^d} \le \frac{1}{n V_d n^{-d/(s+d)}} \lesssim_{s,L,d} n^{-s/(s+d)}. \tag{27}$$

It follows from Lemma 2 that in this case

$$f(x) \quad \lesssim_{s,L,d} \quad f_{R_k(x)}(x) \vee \left( f_{R_k(x)}(x) V_d R_k(x)^d \right)^{s/(s+d)} \tag{28}$$

$$\le \quad n^{-s/(s+d)} \vee n^{-s/(s+d)} = n^{-s/(s+d)}. \tag{29}$$

Hence, we have

$$C_2 \quad = \quad \mathbb{E}\left[ (f_{R_k(x)}(x) - f(x)) \mathbb{1}\left( n^{-1/(s+d)} < R_k(x) \le g(x; f, n) \right) \right] \tag{30}$$

$$\le \quad \mathbb{E}\left[ (f_{R_k(x)}(x) + f(x)) \mathbb{1}\left( n^{-1/(s+d)} < R_k(x) \le g(x; f, n) \right) \right] \tag{31}$$

$$\lesssim_{s,L,d} \quad n^{-s/(s+d)}. \tag{32}$$

3. Term $C_3$: we have

$$C_3 \quad \le \quad \mathbb{E}\left[ (f_{R_k(x)}(x) + f(x)) \mathbb{1}\left( R_k(x) > g(x; f, n) \vee n^{-1/(s+d)} \right) \right]. \tag{33}$$

For any $x$ such that $R_k(x) > n^{-1/(s+d)}$, we have

$$f_{R_k(x)}(x) \quad \lesssim_{s,L,d} \quad V_d R_k(x)^d f_{R_k(x)}(x) n^{d/(s+d)}, \tag{34}$$

and by Lemma 2,

$$f(x) \quad \lesssim_{s,L,d} \quad f_{R_k(x)}(x) \vee (V_d R_k(x)^d f_{R_k(x)}(x))^{s/(s+d)} \tag{35}$$

$$\le \quad f_{R_k(x)}(x) + (V_d R_k(x)^d f_{R_k(x)}(x))^{s/(s+d)}. \tag{36}$$

Hence,

$$f(x) + f_{R_k(x)}(x) \quad \lesssim_{s,L,d} \quad 2 f_{R_k(x)}(x) + (V_d R_k(x)^d f_{R_k(x)}(x))^{s/(s+d)} \tag{37}$$

$$\lesssim_{s,L,d} \quad V_d R_k(x)^d f_{R_k(x)}(x) n^{d/(s+d)} + (V_d R_k(x)^d f_{R_k(x)}(x))^{s/(s+d)} \tag{38}$$

$$\lesssim_{s,L,d} \quad V_d R_k(x)^d f_{R_k(x)}(x) n^{d/(s+d)}, \tag{39}$$

where in the last step we have used the fact that $V_d R_k(x)^d f_{R_k(x)}(x) > n^{-1}$ since $R_k(x) > g(x; f, n)$. Finally, we have

$$C_3 \quad \lesssim_{s,L,d} \quad n^{d/(s+d)} \mathbb{E}[(V_d R_k(x)^d f_{R_k(x)}(x)) \mathbb{1}(R_k(x) > g(x; f, n))] \tag{40}$$

$$= \quad n^{d/(s+d)} \mathbb{E}\left[ (V_d R_k(x)^d f_{R_k(x)}(x)) \mathbb{1}\left( V_d R_k(x)^d f_{R_k(x)}(x) > 1/n \right) \right]. \tag{41}$$

Note that $V_d R_k(x)^d f_{R_k(x)}(x) \sim \mathsf{Beta}(k, n+1-k)$, and if $Y \sim \mathsf{Beta}(k, n+1-k)$, we have

$$\mathbb{E}[Y^2] = \left( \frac{k}{n+1} \right)^2 + \frac{k(n+1-k)}{(n+1)^2(n+2)} \lesssim_k \frac{1}{n^2}. \tag{42}$$

Notice that $\mathbb{E}[Y \mathbb{1}(Y > 1/n)] \le n \mathbb{E}[Y^2]$. Hence, we have

$$C_3 \quad \lesssim_{s,L,d} \quad n^{d/(s+d)} n \, \mathbb{E}\left[ (V_d R_k(x)^d f_{R_k(x)}(x))^2 \right] \tag{43}$$

$$\lesssim_{s,L,d,k} \quad \frac{n^{d/(s+d)} n}{n^2} = n^{-s/(s+d)}. \tag{44}$$

## 2.2 Upper bound on $\mathbb{E}\left[\ln \frac{f(X)}{f_{R_k(X)}(X)}\right]$

By splitting the term into two parts, we have

$$
\mathbb{E}\left[\ln \frac{f(X)}{f_{R_k(X)}(X)}\right] = \mathbb{E}\left[\int_{[0,1]^d \cap \{x:f(x)\neq 0\}} f(x) \ln \frac{f(x)}{f_{R_k(x)}(x)} dx\right] \tag{45}
$$

$$
= \mathbb{E}\left[\int_A f(x) \ln \frac{f(x)}{f_{R_k(x)}(x)} \mathbb{1}(f_{R_k(x)}(x) > n^{-s/(s+d)}) dx\right] \tag{46}
$$

$$
+ \mathbb{E}\left[\int_A f(x) \ln \frac{f(x)}{f_{R_k(x)}(x)} \mathbb{1}(f_{R_k(x)}(x) \leq n^{-s/(s+d)}) dx\right] \tag{47}
$$

$$
= C_4 + C_5. \tag{48}
$$

here we denote $A = [0,1]^d \cap \{x : f(x) \neq 0\}$ for simplicity of notation. For the term $C_4$, we have

$$
C_4 \leq \mathbb{E}\left[\int_A f(x)\left(\frac{f(x) - f_{R_k(x)}(x)}{f_{R_k(x)}(x)}\right)\mathbb{1}(f_{R_k(x)}(x) > n^{-s/(s+d)}) dx\right] \tag{49}
$$

$$
= \mathbb{E}\left[\int_A \frac{(f(x) - f_{R_k(x)}(x))^2}{f_{R_k(x)}(x)}\mathbb{1}(f_{R_k(x)}(x) > n^{-s/(s+d)}) dx\right] \tag{50}
$$

$$
+ \mathbb{E}\left[\int_A \left(f(x) - f_{R_k(x)}(x)\right)\mathbb{1}(f_{R_k(x)}(x) > n^{-s/(s+d)}) dx\right] \tag{51}
$$

$$
\leq n^{s/(s+d)}\mathbb{E}\left[\int_A \left(f(x) - f_{R_k(x)}(x)\right)^2 dx\right] + \mathbb{E}\left[\int_A \left(f(x) - f_{R_k(x)}(x)\right) dx\right]. \tag{52}
$$

In the proof of upper bound of $\mathbb{E}\left[\ln \frac{f_{R_k(X)}(X)}{f(X)}\right]$, we have shown that $\mathbb{E}[f_{R_k(x)}(x) - f(x)] \lesssim_{s,L,d,k} n^{-s/(s+d)}$ for any $x \in A$. Similarly as in the proof of upper bound of $\mathbb{E}\left[\ln \frac{f_{R_k(X)}(X)}{f(X)}\right]$, we have $\mathbb{E}\left[(f_{R_k(x)}(x) - f(x))^2\right] \lesssim_{s,L,d,k} n^{-2s/(s+d)}$ for every $x \in A$. Therefore, we have

$$
C_4 \lesssim_{s,L,d,k} n^{s/(s+d)}n^{-2s/(s+d)} + n^{-s/(s+d)} \lesssim_{s,L,d,k} n^{-s/(s+d)}. \tag{53}
$$

Now we consider $C_5$. We conjecture that $C_5 \lesssim_{s,L,d,k} n^{-s/(s+d)}$ in this case, but we were not able to prove it. Below we prove that $C_5 \lesssim_{s,L,d,k} n^{-s/(s+d)} \ln(n+1)$. Define the function

$$
M(x) = \sup_{t>0} \frac{1}{f_t(x)}. \tag{54}
$$

Since $f_{R_k(x)}(x) \leq n^{-s/(s+d)}$, we have $M(x) = \sup_{t>0}(1/f_t(x)) \geq 1/f_{R_k(x)}(x) \geq n^{s/(s+d)}$. Denote $\ln^+(y) = \max\{\ln(y), 0\}$ for any $y > 0$, therefore, we have that

$$
C_5 \leq \mathbb{E}\left[\int_A f(x)\ln^+\left(\frac{f(x)}{f_{R_k(x)}(x)}\right)\mathbb{1}(f_{R_k(x)}(x) \leq n^{-s/(s+d)}) dx\right] \tag{55}
$$

$$
\leq \mathbb{E}\left[\int_A f(x)\ln^+\left(\frac{f(x)}{f_{R_k(x)}(x)}\right)\mathbb{1}(M(x) \geq n^{s/(s+d)}) dx\right] \tag{56}
$$

$$
\leq \int_A f(x)\mathbb{E}\left[\ln^+\left(\frac{1}{(n+1)V_d R_k(x)^d f_{R_k(x)}(x)}\right)\right]\mathbb{1}(M(x) \geq n^{s/(s+d)}) dx \tag{57}
$$

$$
+ \int_A f(x)\mathbb{E}\left[\ln^+\left((n+1)V_d R_k(x)^d f(x)\right)\right]\mathbb{1}(M(x) \geq n^{s/(s+d)}) dx \tag{58}
$$

$$
= C_{51} + C_{52}, \tag{59}
$$

where the last inequality uses the fact $\ln^+(xy) \le \ln^+ x + \ln^+ y$ for all $x, y > 0$. As for $C_{51}$, since $V_d R_k(x)^d f_{R_k(x)}(x) \sim \text{Beta}(k, n+1-k)$, and for $Y \sim \text{Beta}(k, n+1-k)$, we have

$$\mathbb{E}\left[\ln^+\left(\frac{1}{(n+1)Y}\right)\right] = \int_0^{\frac{1}{n+1}} \ln\left(\frac{1}{(n+1)x}\right) p_Y(x) dx \tag{60}$$

$$= \mathbb{E}\left[\ln\left(\frac{1}{(n+1)Y}\right)\right] + \int_{\frac{1}{n+1}}^1 \ln\left((n+1)x\right) p_Y(x) dx \tag{61}$$

$$\le \mathbb{E}\left[\ln\left(\frac{1}{(n+1)Y}\right)\right] + \ln(n+1) \int_{\frac{1}{n+1}}^1 p_Y(x) dx \tag{62}$$

$$\le \mathbb{E}\left[\ln\left(\frac{1}{(n+1)Y}\right)\right] + \ln(n+1) \tag{63}$$

$$\le \ln(n+1) \tag{64}$$

where in the last inequality we used the fact that $\mathbb{E}\left[\ln\left(\frac{1}{(n+1)Y}\right)\right] = \psi(n+1) - \psi(k) - \ln(n+1) \le 0$ for any $k \ge 1$. Hence,

$$C_{51} \lesssim_{s,L,d} \ln(n+1) \int_A f(x) \mathbb{1}(M(x) \ge n^{s/(s+d)}) dx. \tag{65}$$

Now we introduce the following lemma, which is proved in Appendix C.

**Lemma 3** *Let $\mu_1, \mu_2$ be two Borel measures that are finite on the bounded Borel sets of $\mathbb{R}^d$. Then, for all $t > 0$ and any Borel set $A \subset \mathbb{R}^d$,*

$$\mu_1\left(\left\{x \in A : \sup_{0 < \rho \le D}\left(\frac{\mu_2(B(x,\rho))}{\mu_1(B(x,\rho))}\right) > t\right\}\right) \le \frac{C_d}{t}\mu_2(A_D). \tag{66}$$

*Here $C_d > 0$ is a constant that depends only on the dimension $d$ and*

$$A_D = \{x : \exists y \in A, |y-x| \le D\}. \tag{67}$$

Applying the second part of Lemma 3 with $\mu_2$ being the Lebesgue measure and $\mu_1$ being the measure specified by $f(x)$ on the torus, we can view the function $M(x)$ as

$$M(x) = \sup_{0 < \rho \le 1/2}\frac{\mu_2(B(x,\rho))}{\mu_1(B(x,\rho))}. \tag{68}$$

Taking $A = [0,1]^d \cap \{x : f(x) \ne 0\}, t = n^{s/(s+d)}$, then $\mu_2(A_{\frac{1}{2}}) \le 2^d$, so we know that

$$C_{51} \lesssim_{s,L,d} \ln(n+1) \cdot \int_A f(x)\mathbb{1}(M(x) \ge n^{s/(s+d)}) dx \tag{69}$$

$$= \ln(n+1) \cdot \mu_1\left(x \in [0,1]^d, f(x) \ne 0, M(x) \ge n^{s/(s+d)}\right) \tag{70}$$

$$\le \ln(n+1) \cdot C_d n^{-s/(s+d)}\mu_2(A_{\frac{1}{2}}) \lesssim_{s,L,d} n^{-s/(s+d)}\ln(n+1). \tag{71}$$

Now we deal with $C_{52}$. Recall that in Lemma 2, we know that $f(x) \lesssim_{s,L,d} 1$ for any $x$, and $R_k(x) \le 1$, so $\ln^+((n+1)V_d R_k(x)^d f(x)) \lesssim_{s,L,d} \ln(n+1)$. Therefore,

$$C_{52} \lesssim_{s,L,d} \ln(n+1) \cdot \int_A f(x)\mathbb{1}(M(x) \ge n^{s/(s+d)}) dx \tag{72}$$

$$\lesssim_{s,L,d} n^{-s/(s+d)}\ln(n+1). \tag{73}$$

Therefore, we have proved that $C_5 \le C_{51} + C_{52} \lesssim_{s,L,d} n^{-s/(s+d)}\ln(n+1)$, which completes the proof of the upper bound on $\mathbb{E}\left[\ln\frac{f(X)}{f_{R_k(X)}(X)}\right]$.

## 3  Future directions

It is an tempting question to ask whether one can close the logarithmic gap between Theorem 1 and 2. We believe that neither the upper bound nor the lower bound are tight. In fact, we conjecture that the upper bound in Theorem 1 could be improved to $n^{-\frac{s}{s+d}} + n^{-1/2}$ due to a more careful analysis of the bias, since Hardy–Littlewood maximal inequalities apply to arbitrary measurable functions but we have assumed regularity properties of the underlying density. We conjecture that the minimax lower bound could be improved to $(n \ln n)^{-\frac{s}{s+d}} + n^{-1/2}$, since a kernel density estimator based differential entropy estimator was constructed in [26] which achieves upper bound $(n \ln n)^{-\frac{s}{s+d}} + n^{-1/2}$ over $\mathcal{H}_d^s(L; [0,1]^d)$ with the knowledge of $s$.

It would be interesting to extend our analysis to that of the $k$-nearest neighbor based Kullback–Leibler divergence estimator [59]. The discrete case has been studied recently [28, 7].

It is also interesting to analyze $k$-nearest neighbor based mutual information estimators, such as the KSG estimator [34], and show that they are "near"-optimal and adaptive to both the smoothness and the dimension of the distributions. There exists some analysis of the KSG estimator [21] but we suspect the upper bound is not tight. Moreover, a slightly revised version of KSG estimator is proved to be consistent even if the underlying distribution is not purely continuous nor purely discrete [19], but the optimality properties are not yet well understood.

## Footnotes

[1]Precisely, we have $\int_{B(X_i, R_{i,k})} f(u) du \sim \text{Beta}(k, n-k)$ [4, Chap. 1.2]. A $\text{Beta}(k, n-k)$ distributed random variable has mean $\frac{k}{n}$.

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
