[Supplementary Material]

## A  Definition of Hölder Ball

In order to define the Hölder ball in the unit cube $[0, 1]^d$, we first review the definition of Hölder ball in $\mathbb{R}^d$.

**Definition 1 (Hölder ball in $\mathbb{R}^d$)** *The Hölder ball $\mathcal{H}_d^s(L; \mathbb{R}^d)$ is specified by the parameters $s > 0$ (order of smoothness), $d \in \mathbb{Z}_+$ (dimension of the argument) and $L > 0$ (smoothness constant) and is as follows. A positive real $s$ can be uniquely represented as*

$$s = m + \alpha, \tag{74}$$

*where $m$ is a nonnegative integer and $0 < \alpha \leq 1$. By definition, $\mathcal{H}_d^s(L; \mathbb{R}^d)$ is comprised of all $m$ times continuously differentiable functions*

$$f : \mathbb{R}^d \mapsto \mathbb{R}, \tag{75}$$

*with Hölder continuous, with exponent $\alpha$ and constant $L$, derivatives of order $m$:*

$$\|D^m f(x)[\delta_1, \ldots, \delta_m] - D^m f(x')[\delta_1, \ldots, \delta_m]\| \leq L\|x - x'\|^\alpha \|\delta\|^m, \quad \forall x, x' \in \mathbb{R}^d, \delta \in \mathbb{R}^d. \tag{76}$$

*Here $\|\cdot\|$ is the Euclidean norm on $\mathbb{R}^d$, and $D^m f(x)[\delta_1, \ldots, \delta_m]$ is the $m$-th differential of $f$ taken as a point $x$ along the directions $\delta_1, \ldots, \delta_m$:*

$$D^m f(x)[\delta_1, \ldots, \delta_m] \quad = \frac{\partial^m}{\partial t_1 \ldots \partial t_m}\bigg|_{t_1 = t_2 = \ldots = t_m = 0} f(x + t_1\delta_1 + \ldots + t_m\delta_m). \tag{77}$$

In this paper, we consider functions that lie in Hölder balls in $[0, 1]^d$. The Hölder ball in the compact set $[0, 1]^d$ is defined as follows.

**Definition 2 (Hölder ball in the unit cube)** *A function $f : [0, 1]^d \mapsto \mathbb{R}$ is said to belong to the Hölder ball $\mathcal{H}_d^s(L; [0, 1]^d)$ if and only if there exists another function $f_1 \in \mathcal{H}_d^s(L; \mathbb{R}^d)$ such that*

$$f(x) \quad = f_1(x), \quad x \in [0, 1], \tag{78}$$

*and $f_1(x)$ is a 1-periodic function in each variable. Here $\mathcal{H}_d^s(L; [0, 1]^d)$ is introduced in Definition 1. In other words,*

$$f_1(x + e_j) \quad = f_1(x), \quad \forall x \in \mathbb{R}^d, 1 \leq j \leq d, \tag{79}$$

*where $\{e_j : 1 \leq j \leq d\}$ is the standard basis in $\mathbb{R}^d$.*

Definition 2 has appeared in the literature [35]. It is motivated by the observations that sliding window kernel methods usually can not deal with the boundary effects without additional assumptions [31]. Indeed, near the boundary the sliding window kernel density estimator may have a significantly larger bias than that of the interior points. In the nonparametric statistics literature, it is usually assumed that the density has its value and all the derivatives vanishing at the boundary, which is stronger than our assumptions.

## B  Variance upper bound in Theorem 1

Our goal is to prove

$$\mathsf{Var}\left(\hat{h}_{n,k}(\mathbf{X})\right) \lesssim_{d,k} \frac{1}{n}. \tag{80}$$

The proof is based on the analysis in [4, Section 7.2] which utilizes the Efron–Stein inequality. Let $\mathbf{X}^{(i)} = \{X_1, \ldots, X_{i-1}, X_i', X_{i+1}, \ldots, X_n\}$ be a set of sample where only $X_i$ is replaced by $X_i'$. Then Efron–Stein inequality [13] states

$$\mathsf{Var}\left(\hat{h}_{n,k}(\mathbf{X})\right) \quad \leq \quad \frac{1}{2} \sum_{i=1}^{n} \mathbb{E}\left[\left(\hat{h}_{n,k}(\mathbf{X}) - \hat{h}_{n,k}(\mathbf{X}^{(i)})\right)^2\right] \tag{81}$$

Note that KL estimator is symmetric of sample indices, so $\hat{h}_{n,k}(\mathbf{X}) - \hat{h}_{n,k}(\mathbf{X}^{(i)})$ has the same distribution for any $i$. Furthermore, we bridge $\hat{h}_{n,k}(\mathbf{X})$ and $\hat{h}_{n,k}(\mathbf{X}^{(i)})$ by introducing an estimator from $n-1$ samples. Precisely, for any $i = 2, \ldots, n$, define $R'_{i,k}$ be the $k$-nearest neighbor distance from $X_i$ to $\{X_2, \ldots, X_n\}$ (note that $X_1$ is removed), under the distance $d(x,y) = \min_{m \in \mathbb{Z}^d} \|x - y - m\|$. Define

$$\hat{h}_{n-1,k}(\mathbf{X}) = -\psi(k) + \frac{1}{n}\sum_{i=2}^{n} \ln(n\lambda(B(X_i, R'_{i,k}))) \,. \tag{82}$$

Notice that $\hat{h}_{n,k}(\mathbf{X}) - \hat{h}_{n-1,k}(\mathbf{X})$ has the same distribution as $\hat{h}_{n,k}(\mathbf{X}^{(1)}) - \hat{h}_{n-1,k}(\mathbf{X})$. Therefore, the variance is bounded by

$$\begin{aligned}
\mathsf{Var}\left(\hat{h}_{n,k}(\mathbf{X})\right) &\leq \frac{n}{2}\mathbb{E}\left[\left(\hat{h}_{n,k}(\mathbf{X}) - \hat{h}_{n,k}(\mathbf{X}^{(1)})\right)^2\right] \\
&= 2n\mathbb{E}\left[\left(\hat{h}_{n,k}(\mathbf{X}) - \hat{h}_{n-1,k}(\mathbf{X})\right)^2\right]
\end{aligned} \tag{83}$$

Now we deal with the term $\mathbb{E}\left[\left(\hat{h}_{n,k}(\mathbf{X}) - \hat{h}_{n-1,k}(\mathbf{X})\right)^2\right]$. Define the indicator function

$$E_i^{(k)} = \mathbb{I}\{X_1 \text{ is in the } k-\text{nearest neighbor of } X_i\}. \tag{84}$$

for $i \neq 1$. Note that $R'_{i,k} = R_{i,k}$ if $E_i^{(k)} \neq 1$ and $i \neq 1$. As shown in [19, Lemma B.1], the set $S = \{i : E_i^{(k)} = 1\}$ has cardinality at most $k\beta_d$ for a constant $\beta_d$ only depends on $d$. Therefore, we have

$$\begin{aligned}
&\mathsf{Var}\left(\hat{h}_{n,k}(\mathbf{X})\right) \\
&\leq 2n\mathbb{E}\left[\left(\hat{h}_{n,k}(\mathbf{X}) - \hat{h}_{n-1,k}(\mathbf{X})\right)^2\right] \tag{85} \\
&= 2n\mathbb{E}\left[\frac{1}{n^2}\left(\sum_{i \in S \cup \{1\}} \ln(n\lambda(B(X_i, R_{i,k}))) - \sum_{i \in S} \ln(n\lambda(B(X_i, R'_{i,k})))\right)^2\right] \tag{86} \\
&\leq \frac{2}{n}\mathbb{E}\left[(1 + 2|S|)\left(\sum_{i \in S \cup \{1\}} \ln^2(n\lambda(B(X_i, R_{i,k}))) + \sum_{i \in S} \ln^2(n\lambda(B(X_i, R'_{i,k})))\right)\right] \tag{87} \\
&\lesssim_{d,k} \frac{1}{n}\left(\mathbb{E}\left[\ln^2(n\lambda(B(X_1, R_{1,k})))\right] + \mathbb{E}\left[\ln^2(n\lambda(B(X_1, R'_{1,k})))\right]\right) \,. \tag{88}
\end{aligned}$$

Now we prove that $\mathbb{E}\left[\ln^2(n\lambda(B(X_1, R_{1,k})))\right] \lesssim_{d,k} 1$ and $\mathbb{E}\left[\ln^2(n\lambda(B(X_1, R'_{1,k})))\right] \lesssim_{d,k} 1$. Using Cauchy-Schwarz inequality, we have

$$\begin{aligned}
&\mathbb{E}\left[\ln^2(n\lambda(B(X_1, R_{1,k})))\right] \\
&\leq 2\left(\mathbb{E}\left[\ln^2(\frac{\lambda(B(X_1, R_{1,k}))}{\mu(B(X_1, R_{1,k}))})\right] + \mathbb{E}\left[\ln^2(n\mu(B(X_1, R_{1,k})))\right]\right) \,, \tag{89} \\
&\mathbb{E}\left[\ln^2(n\lambda(B(X_1, R'_{1,k})))\right] \\
&\leq 3\left(\mathbb{E}\left[\ln^2(\frac{\lambda(B(X_1, R'_{1,k}))}{\mu(B(X_1, R'_{1,k}))})\right] + \mathbb{E}\left[\ln^2((n-1)\mu(B(X_1, R'_{1,k})))\right] + \ln^2(\frac{n}{n-1})\right) \,. \tag{90}
\end{aligned}$$

Since $\mu(B(X_1, R_{1,k})) \sim \mathsf{Beta}(k, n+1-k)$ and $\mu(B(X_1, R'_{1,k})) \sim \mathsf{Beta}(k, n-k)$, therefore we know that both $\mathbb{E}\left[\ln^2(n\mu(B(X_1, R_{1,k})))\right]$ and $\mathbb{E}\left[\ln^2((n-1)\mu(B(X_1, R'_{1,k})))\right]$ equal to certain constants that only depends on $k$. $\ln^2(n/(n-1))$ is smaller than $\ln^2 2$ for $n \geq 2$. So we only need

to prove that $\mathbb{E}\left[\ln^2\left(\frac{\lambda(B(X_1,R_{1,k}))}{\mu(B(X_1,R_{1,k}))}\right)\right] \lesssim_{d,k} 1$ and $\mathbb{E}\left[\ln^2\left(\frac{\lambda(B(X_1,R'_{1,k}))}{\mu(B(X_1,R'_{1,k}))}\right)\right] \lesssim_{d,k} 1$. Recall that we have defined the maximal function as follows,

$$M(x) = \sup_{0 \leq r \leq 1/2} \frac{\lambda(B(x,r))}{\mu(B(x,r))} . \tag{91}$$

Similarly, we define

$$m(x) = \sup_{0 \leq r \leq 1/2} \frac{\mu(B(x,r))}{\lambda(B(x,r))} . \tag{92}$$

Therefore,

$$\mathbb{E}\left[\ln^2\left(\frac{\lambda(B(X_1,R_{1,k}))}{\mu(B(X_1,R_{1,k}))}\right)\right] \leq \mathbb{E}\left[\max\{\ln^2(M(x)), \ln^2(m(x))\}\right] \tag{93}$$

$$\leq \mathbb{E}\left[\ln^2(M(x)+1) + \ln^2(m(x)+1)\right] \tag{94}$$

$$= \mathbb{E}\left[\ln^2(M(x)+1)\right] + \mathbb{E}\left[\ln^2(m(x)+1)\right] . \tag{95}$$

Similarly this inequality holds if we replace $R_{1,k}$ by $R'_{1,k}$. By Lemma 3, we have

$$\mathbb{E}\left[\ln^2(M(x)+1)\right] = \int_{[0,1]^d} \ln^2(M(x)+1)d\mu(x) \tag{96}$$

$$= \int_{t=0}^{\infty} \mu\left(\{x \in [0,1]^d : \ln^2(M(x)+1) > t\}\right) dt \tag{97}$$

$$= \int_{t=0}^{\infty} \mu\left(\left\{x \in [0,1]^d : M(x) > e^{\sqrt{t}} - 1\right\}\right) dt \tag{98}$$

$$\lesssim_d \int_{t=0}^{\infty} \frac{1}{e^{\sqrt{t}}-1} dt \lesssim_d 1. \tag{99}$$

For $\mathbb{E}[\ln^2(m(x)+1)]$, we rewrite the term as

$$\mathbb{E}\left[\ln^2(m(x)+1)\right] = \int_{[0,1]^d} f(x)\ln^2(m(x)+1)d\lambda(x) \tag{100}$$

$$= \int_{t=0}^{\infty} \lambda\left(\{x \in [0,1]^d : f(x)\ln^2(m(x)+1) > t\}\right) dt . \tag{101}$$

For $t \leq 100$, simply we use $\lambda\left(\{x \in [0,1]^d : f(x)\ln^2(m(x)+1) > t\}\right) \leq 1$. For $t > 100$, $f(x)\ln^2(m(x)+1) > t$ implies either $m(x) > t^2$ or $f(x) > t/\ln^2(t^2+1)$. Moreover, if $f(x) > t/\ln^2(t^2+1)$ then

$$f(x)\ln^2 f(x) > \frac{t(\ln t - 2\ln\ln(t^2+1))^2}{\ln^2(t^2+1)} > \frac{t}{10000} \tag{102}$$

since $(\ln t - 2\ln\ln(t^2+1))^2/\ln^2(t^2+1) > 1/10000$ for any $t > 100$. So for $t > 100$,

$$\lambda\left(\{x \in [0,1]^d : f(x)\ln^2(m(x)+1) > t\}\right)$$
$$\leq \lambda\left(\{x \in [0,1]^d : m(x) > t^2\}\right) + \lambda\left(\{x \in [0,1]^d : f(x)\ln^2 f(x) > t/10000\}\right) . \tag{103}$$

Therefore,

$$\int_{t=0}^{\infty} \lambda\left(\{x \in [0,1]^d : f(x)\ln^2(m(x)+1) > t\}\right) dt \tag{104}$$

$$\leq \int_{t=0}^{100} 1\, dt + \int_{t=100}^{\infty} \lambda\left(\{x \in [0,1]^d : m(x) > t^2\}\right) dt$$

$$+ \int_{t=100}^{\infty} \lambda\left(\{x \in [0,1]^d : f(x)\ln^2 f(x) > t/10000\}\right) dt \tag{105}$$

$$\lesssim_d 100 + \int_{t=100}^{\infty} \frac{1}{t^2} dt + 10000 \int_{[0,1]^d} f(x)\ln^2 f(x)dx \tag{106}$$

$$\lesssim 1. \tag{107}$$

Hence, the proof is completed.

# C Proof of lemmas

In this section we provide proofs of lemmas used in the paper.

## C.1 Proof of Lemma 1

We consider the cases $s \in (0, 1]$ and $s \in (1, 2]$ separately. For $s \in (0, 1]$, following the definition of Hölder smoothness, we have,

$$| f_t(x) - f(x) | \quad = \quad \left| \frac{1}{V_d t^d} \int_{u:||u-x|| \leq t} f(u) du - f(x) \right| \tag{108}$$

$$\leq \quad \frac{1}{V_d t^d} \int_{u:||u-x|| \leq t} |f(u) - f(x)| du \tag{109}$$

$$\leq \quad \frac{1}{V_d t^d} \int_{u:||u-x|| \leq t} L\|u - x\|^s du . \tag{110}$$

By denoting $\rho = \|u - x\|$ and considering $\theta \in S^{d-1}$ on the unit $d$-dimensional sphere, we rewrite the above integral using polar coordinate system and obtain,

$$| f_t(x) - f(x) | \quad \leq \quad \frac{1}{V_d t^d} \int_{\rho=0}^{t} \int_{\theta \in S^{d-1}} L\rho^s \rho^{d-1} d\rho d\theta \tag{111}$$

$$= \quad \frac{1}{V_d t^d} \int_{\rho=0}^{t} dV_d L\rho^{s+d-1} d\rho \tag{112}$$

$$= \quad \frac{dV_d L t^{s+d}}{(s+d)V_d t^d} = \frac{dL t^s}{s+d} . \tag{113}$$

Now we consider the case $s \in (1, 2]$. Now we rewrite the difference as

$$| f_t(x) - f(x) | \quad = \quad \left| \frac{1}{V_d t^d} \int_{u:\|u-x\| \leq t} f(u) du - f(x) \right| \tag{114}$$

$$= \quad \left| \frac{1}{2V_d t^d} \int_{v:\|v\| \leq t} ( f(x+v) + f(x-v) ) dv - f(x) \right| \tag{115}$$

$$\leq \quad \frac{1}{2V_d t^d} \int_{v:\|v\| \leq t} \left| f(x+v) + f(x-v) - 2f(x) \right| dv . \tag{116}$$

For fixed $v$, we bound $|f(x+v) + f(x-v) - 2f(x)|$ using the Gradient Theorem and the definition of Hölder smoothness as follows,

$$|f(x + v) + f(x - v) - 2f(x)| \tag{117}$$

$$= \quad \left| ( f(x+v) - f(x) ) + ( f(x-v) - f(x) ) \right| \tag{118}$$

$$= \quad \left| \int_{\alpha=0}^{1} \nabla f(x+\alpha v) \cdot d(x+\alpha v) + \int_{\alpha=0}^{-1} \nabla f(x+\alpha v) \cdot d(x+\alpha v) \right| \tag{119}$$

$$= \quad \left| \int_{\alpha=0}^{1} ( \nabla f(x+\alpha v) \cdot v ) d\alpha - \int_{\alpha=0}^{1} ( \nabla f(x-\alpha v) \cdot v ) d\alpha \right| \tag{120}$$

$$= \quad \left| \int_{\alpha=0}^{1} ( \nabla f(x+\alpha v) - \nabla f(x-\alpha v) ) \cdot v d\alpha \right| \tag{121}$$

$$\leq \quad \int_{\alpha=0}^{1} \| \nabla f(x+\alpha v) - \nabla f(x-\alpha v)\| \|v\| d\alpha \tag{122}$$

$$\leq \quad \int_{\alpha=0}^{1} L\|2\alpha v\|^{s-1} \|v\| d\alpha \tag{123}$$

$$= \quad L\|v\|^s \int_0^1 (2\alpha)^{s-1} d\alpha = \frac{L\|v\|^s 2^{s-1}}{s} . \tag{124}$$

Plug it into (116) and using the similar method in the $s \in (0, 1]$ case, we have

$$|f_t(x) - f(x)| \quad \leq \quad \frac{1}{2V_d t^d} \int_{v:\|v\| \leq t} \frac{L\|v\|^s 2^{s-1}}{s} dv \tag{125}$$

$$= \quad \frac{1}{2V_d t^d} \int_{\rho=0}^{t} \int_{\theta \in S^{d-1}} \frac{L\rho^s 2^{s-1}}{s} \rho^{d-1} d\rho d\theta \tag{126}$$

$$= \quad \frac{1}{2V_d t^d} \int_{\rho=0}^{t} \frac{dV_d L \rho^{s+d-1} 2^{s-1}}{s} d\rho \tag{127}$$

$$= \quad \frac{1}{2V_d t^d} \frac{dV_d L 2^{s-1}}{s} \frac{t^{s+d}}{s+d} \leq \frac{dLt^s}{s+d} , \tag{128}$$

where the last inequality uses the fact that $s \in (1, 2]$.

## C.2 Proof of Lemma 2

We consider the following two cases. If $f(x) \geq 2dLt^s/(s+d)$, then by Lemma 1, we have

$$f(x) \leq f_t(x) + \frac{dLt^s}{s+d} \leq f_t(x) + \frac{f(x)}{2} . \tag{129}$$

Hence, $f(x) \leq 2f_t(x)$ in this case. If $f(x) < 2dLt^s/(s+d)$, then define $t_0 = (f(x)(s+d)/2dL)^{1/s} < t$. By the nonnegativity of $f$, we have

$$f_t(x) V_d t^d \tag{130}$$

$$= \quad \int_{B(x,t)} f(x)dx \geq \int_{B(x,t_0)} f(x)dx = f_{t_0}(x) V_d t_0^d \tag{131}$$

$$\geq \quad \left( f(x) - \frac{dLt_0^s}{s+d} \right) V_d t_0^d \tag{132}$$

$$= \quad f(x) V_d \left( \frac{f(x)(s+d)}{2dL} \right)^{d/s} - \frac{dL}{s+d} V_d \left( \frac{f(x)(s+d)}{2dL} \right)^{(s+d)/s} \tag{133}$$

$$= \quad f(x)^{(s+d)/s} V_d \left( \frac{s+d}{dL} \right)^{d/s} \left( 2^{-d/s} - 2^{-(s+d)/s} \right) . \tag{134}$$

Therefore, we have $f(x) \lesssim_{s,L,d} (f_t(x) V_d t^d)^{s/(s+d)}$ in this case. We obtain the desired statement by combining the two cases. Furthermore, by taking $t = 1/2$, we have $V_d t^d f_t(x) < 1$, so $f_t(x) \lesssim_{s,L,d} 1$. By applying this lemma immediately we obtain $f(x) \lesssim_{s,L,d} 1$.

## C.3 Proof of Lemma 3

We first introduce the Besicovitch covering lemma, which plays a crucial role in the analysis of nearest neighbor methods.

**Lemma 4** *[15, Theorem 1.27][Besicovitch covering lemma] Let $A \subset \mathbb{R}^d$, and suppose that $\{B_x\}_{x \in A}$ is a collection of balls such that $B_x = B(x, r_x), r_x > 0$. Assume that $A$ is bounded or that $\sup_{x \in A} r_x < \infty$. Then there exist an at most countable collection of balls $\{B_j\}$ and a constant $C_d$ depending only on the dimension $d$ such that*

$$A \subset \bigcup_j B_j , \quad \text{and} \quad \sum_j \chi_{B_j}(x) \leq C_d. \tag{135}$$

*Here $\chi_B(x) = \mathbb{1}(x \in B)$.*

Now we are ready to prove the lemma. Let

$$M(x) = \sup_{0 < \rho \leq D} \left( \frac{\mu_2(B(x,\rho))}{\mu_1(B(x,\rho))} \right) . \tag{136}$$

Let $O_t = \{x \in A : M(x) > t\}$. Hence, for all $x \in O_t$, there exists $B_x = B(x, r_x)$ such that $\mu_2(B_x) > t\mu_1(B_x), 0 < r_x \leq D$. It follows from the Besicovitch lemma applying to the set $O_t$ that there exists a set $E \subset O_t$, which has at most countable cardinality, such that

$$O_t \subset \bigcup_{j \in E} B_j, \quad \text{and} \quad \sum_{j \in E} \chi_{B_j}(x) \leq C_d. \tag{137}$$

Let $A_D = \{x : \exists y \in A, |y - x| \leq D\}$, therefore $B_j \subset A_D$ for every $j$. Then,

$$
\begin{aligned}
\mu_1(O_t) &\leq \sum_{j \in E} \mu_1(B_j) < \frac{1}{t} \sum_{j \in E} \mu_2(B_j) \\
&= \frac{1}{t} \sum_{j \in E} \int_{A_D} \chi_{B_j} d\mu_2 = \frac{1}{t} \int_{A_D} \sum_{j \in E} \chi_{B_j} d\mu_2 \leq \frac{C_d}{t} \mu_2(A_D).
\end{aligned} \tag{138}
$$