[Reviews · NeurIPS 2018]

Reviewer 1



This paper studies the nearest neighbor information estimator, aka the Kozachenko-Leonenko (KL) estimator for the differential entropy. Matching upper and lower bounds (up to log factor) are proven under a H\"older ball condition. The paper is of high quality and clarity. The introductory and the technical parts are smoothly written, although I could not verify all the details of the proofs. References to the literature look great. Intuitions are given along the way of the presentation. There are two significant theoretical contributions. First, estimating the entropy becomes significantly harder when the density is allowed to be close to zero. A previous work I am aware of that handles this situation is [Han, Jiao, Weissman, Wu 2017] that uses sophisticated kernel and polynomial approximation techniques, while this works shows that the simple KL estimator achieves similar effect (albeit under other different assumptions). Second, the KL estimator does not use the smoothness parameter $s$, so it is naturally adaptive and achieves bounds depending on $s$. A couple of minor comments are in the sequel. When introducing the quantity $R_{i,k}$, it should be clearly stated what distance between $X_i$ and its $k$-nearest neighbor" means here. This might be too much beyond the scope of the paper, but do you have results for the KL estimator under the more general Lipschitz ball condition [HJWW17]?

Reviewer 2



Paper 1614 This paper studies the Kozachenko-Leonenko estimator for the differential entropy of a multivariate smooth density that satisfy a periodic boundary condition; an equivalent way to state the condition is to let the density be defined on the [0,1]^d-torus. The authors show that the K-L estimator achieves a rate of convergence that is optimal up to poly-log factors. The result is interesting and the paper is well-written. I could not check the entirety of the proof but the parts I checked are correct. I recommend that the paper be accepted. Some questions and remarks: * The periodic boundary condition is unnatural, though one can imagine that the problem is much harder without it. Can the authors comment on whether anything can be shown for densities over R^d? * Can the weighted KL estimator proposed in [1] be used to control the bias and handle cases where s > 2 ? * It seems that the Besicovitch (typo on line 99) covering lemma implies the Hardy-Littlewood maximal inequality. I only see the variance bound use the former so perhaps the latter result need not be mentioned. On an unrelated note, I am curious whether the authors know what the dependence of the C_d constant on dimensionality d is in the Besicovitch covering lemma. * Can similar estimator and analysis be used for estimating the Renyi entropy? [1] T. Berrett, R. Samworth, and M. Yuan, Efficient multivariate entropy estimation via nearest k-neighbor estimators.

Reviewer 3



This paper shows that the Kozachenko–Leonenko estimator for differential entropy is near optimal considering the minimax risk of differential entropy estimation. In particular, the assumption for density functions to be away from zero is not used, while previous methods use the assumption for the convergence proof. The paper shows a thorough literature review and explain what is the contribution of this paper very clearly. Many parts of the paper are based on the book of Biau and Devroye [4], and the proof explanation in the Appendix covers the fundamental parts of the derivation in the main paper very well. The main result of this paper is also interesting. The Kozachenko–Leonenko estimator and the Kullback-Leibler divergence estimator based on this estimator is known to work well in practice compared with other methods even though nearest neighbor methods are usually considered as a simple but poor method. The derived bound of the error is interestingly has a similar form to the minimum error of such estimators (minimax bound), and the bound is very close to the minimum error as well. This makes reading the paper enjoyable.